# HIV-related challenges and women's self-response: A qualitative study with women living with HIV in Indonesia

**Nelsensius Klau Fauk**[1,2], **Hailay Abrha Gesesew**[1,3]*, **Lillian Mwanri**[1], **Karen Hawke**[4], **Paul Russell Ward**[1]

1 Research Centre for Public Health, Equity and Human Flourishing, Torrens University Australia, Adelaide, South Australia, Australia, 2 Institute of Resource Governance and Social Change, Kupang, Nusa Tenggara Timur, Indonesia, 3 College of Health Sciences, Mekelle University, Mekelle, Tigray, Ethiopia, 4 Infectious Disease—Aboriginal Health, South Australian Health and Medical Research Institute, Adelaide, Australia

☯ These authors contributed equally to this work.

* hailushepi@gmail.com

**Data Availability Statement:** The data relevant to this study cannot be shared publicly as they contain sensitive and potentially identifying participant information. These restrictions were

## Abstract

HIV infection is a major public health concern, with a range of negative impacts on People Living with HIV (PLHIV). A qualitative study in Yogyakarta, Indonesia, using in-depth interviews with 26 Women Living with HIV (WLHIV) was conducted to understand HIV risk factors and impact and their access to HIV care services. This paper describes the self-response of WLHIV towards negative HIV-related experiences facing them and adds to the existing literature which tends to focus on HIV impact only, as opposed to strategies that many WLHIV have used to empower and educate themselves and their family/community. Participants were recruited using the snowball sampling technique. Data analysis was guided by a qualitative data analysis framework. Our study highlighted that WLHIV experienced psychological challenges, stigma and discrimination. However, they demonstrated remarkable self-response and capacity in pursuing effective strategies and support to protect themselves, and educating themselves and others around them to rebuild trust and regain respect and acceptance. Our findings indicate that the needs of WLHIV should be addressed through policy and practice to help them cope with HIV-related psychological and social challenges effectively. Family and community members seem to play an important role in those negative challenges against WLHIV, thus there is also a need for HIV education programs for family and community members to enhance their HIV-health literacy and acceptance of PLHIV.

## Introduction

HIV diagnoses among women in Indonesia have gradually increased over the past five years. The recent Indonesian Ministry of Health report for HIV/AIDS shows the annual number of women newly diagnosed with HIV has increased from 12,573 in 2015 to 17,839 in 2021 [1]. The same overall increasing trend has also been reported for Yogyakarta, the setting of the

imposed by the Social and Behavioural Research Ethics Committee, Flinders University, South Australia, Australia. For more information regarding the ethical approval and for data access requests, please contact the Executive Officer at human.researchethics@flinders.edu.au.

**Funding:** The author(s) received no specific funding for this work.

**Competing interests:** The authors have declared that no competing interests exist.

current study [2]. Women are one of the high-risk groups for HIV infection globally and in Indonesia, and account for the highest proportion of people living with AIDS currently [1, 3]. Women Living with HIV (WLHIV) are therefore highly vulnerable to the detrimental impact of the HIV epidemic, as living with HIV has many interrelated issues, such as psychological challenges and stigma and discrimination reflected in rejection, social exclusion and isolation within families, communities, healthcare facilities and workplace settings [4–6].

Psychological challenges, including stress, anxiety, depression, sadness and embarrassment are common negative impacts faced by WLHIV following their HIV diagnosis [6–9]. The stressors for such psychological challenges on women include the advanced stages of their HIV infection, weak physical condition, and the fear of a breach of confidentiality about their HIV status which may cause shame for their families [6, 8]. Other stressors for these psychological challenges are the women's fear of HIV transmission to their unborn babies, concerns about their children's future, especially in the event of an untimely death, and a lack of resources needed to support their children and family without the additional strain of an HIV diagnosis [4, 10, 11]. Similarly, a lack of social support from others, the experience of social rejection and social isolation, the perceived stigma [8, 12, 13]and poor economic conditions [11, 14] are also reported as determinants of depression, fear and worry among WLHIV.

Stigma and discrimination are also common negative HIV impacts experienced by WLHIV. For example, WLHIV have been reported to experience stigma and discrimination within their family reflected in blame, verbal insults, avoidance, and rejection [5, 8], separation of their personal items from those of other family members, and exclusion from usual family activities [5, 8, 15]. Other studies have reported that WLHIV experience stigma and discrimination within their communities, manifested in social isolation such as rejection and neglect by friends and relatives, and being denied to enter or asked to leave the homes of neighbours [7, 16]. Refusal to share food and drink by neighbours and relatives has also been reported [4, 5], as well as physical assault, harassment, negative labelling, and verbal abuse using discriminatory words and insults by community members [15, 17]. The most commonly reported reason for these discriminatory and stigmatising attitudes and behaviours displayed by family and community is fear of being infected with HIV by interacting with WLHIV [7, 16, 18]. HIV-related stigma and discrimination also occur within healthcare settings, often with WLHIV experiencing criticism, blame, rejection, unnecessary referrals, or being neglected and left untreated by healthcare professionals, again due to fear of being infected with HIV themselves [8, 10, 16, 19].

Though there are many studies on psychological challenges and HIV-related stigma among WLHIV globally, the majority tend to focus on HIV impact only and present them as victims and vectors of HIV infection, rather than examining strategies or responses they have used over their lives [4, 18, 20]. As part of a larger study that explored HIV-risk factors and impacts among WLHIV and their access to HIV care services in Yogyakarta, Indonesia, this paper aims to deeply explore these women's self-response to psychological challenges and HIV stigma facing them following their HIV diagnosis. Although both women and men living with HIV have been reported to experience HIV-related psychological and social challenges, this report focuses on women due to the consideration, as reported in some previous studies in developing countries, including Indonesia, that WLHIV often experience a double burden due to the HIV diagnosis [4, 21]. Regardless of how they acquire the infection (e.g., through their spouses or partners) many WLHIV are often negatively labelled as 'naughty women' or women who have sex with multiple men, female sex workers, disloyal women, etc., which are not experienced by men living with HIV [4, 21]. Narratives of the women in this study were powerful and expressed experiencing psychological pressures, stigma and discrimination by their family members and other people around them following the HIV diagnosis. Despite

experiencing these challenges, they demonstrated great self-response and capacity to pursue effective strategies and supports to protect themselves, and educate others around them to rebuild trust and respect. It is crucial to understand the lived experience of WLHIV themselves to effectively address the impact of the HIV epidemic at multiple levels. This may include government legislation and policy change and immediate healthcare and mental health support to developing evidence-based programs and interventions that address the needs of WLHIV and increase their resilience and capacity to access these supports.

## Methods

### Study setting

Yogyakarta city, the only municipality in the Special Region of Yogyakarta province, Indonesia, and the current study focal setting, is divided into 14 sub-districts and 45 villages [22]. It is inhabited by 636,660 people, with a population density of 13,340 people per km$^2$ [22]. It has two government hospitals and 18 private hospitals, 18 public health centres and nine sub-public health centres. Of the hospitals and community health centres, four hospitals and 10 community health centres provide HIV/AIDS-related health services for 1,353 People Living with HIV (PLHIV) in 2021 [1, 23]. The services include providing health information on HIV and related services, HIV counselling and testing, the cluster of differentiation 4 (CD4) and viral load testing, and antiretroviral therapy (ART) and its adherence measures. HIV-related health information services are provided through workshops for PLHIV, regular focus group discussions and peer support group meetings attended by PLHIV and their companions, nurses and doctors. To the best of our knowledge, there have never been studies that focused on exploring the self-response and capacity of PLHIV in Indonesia, and due to feasibility, familiarity and the potential to successfully conduct this study, Yogyakarta was chosen as the study setting.

### Study design and data collection

This was a qualitative inquiry [24, 25]that was carried out with WLHIV in Yogyakarta, Indonesia. After soliciting permission from an HIV clinic in the study setting, the field researcher solicited the help of the receptionist at the clinic who agreed to post study information sheets on the clinic's information board for potential participants (WLHIV) who accessed HIV care services in the clinic. Potential participants who called and confirmed their willingness to participate in the study were asked to recommend a preferred time and place for an interview. Initial participants who had been interviewed were also asked to further distribute the study information sheets to their eligible colleagues who might be willing to participate. The recruitment was guided by the inclusion criteria that one had to be HIV-positive and aged 18 years or above. The recruitment process took six months, with 26 WLHIV participating in the study.

Participants' age ranged from 20 to 49 years, with the majority between 30 and 49 years. Half of the women were ever married, and the remaining were non-married (divorced, widowed or single/never married). Most participants had been living with HIV between one to five years, and a few had been diagnosed with HIV six to 15 years ago. Several participants reported being diagnosed with other sexually transmitted infections, such as herpes, candidiasis, syphilis, and gonorrhoea. Four participants had also been diagnosed with tuberculosis. Education backgrounds varied, with 13 women graduating from senior high school, six from university, six from junior high school, and one from elementary school. Ten women reported being housewives. The others, except one unemployed, engaged in different kinds of professions as presented in Table 1.

This study was carried out from June to December 2019. In-depth interviews were used to explore participants' perceptions of factors leading to HIV infection, their lived experiences of

**Table 1. Characteristics of the participants.**

| Characteristics | Women living with HIV (n = 26) |
|---|---|
| **Age** | |
| 20–29 | 6 |
| 30–39 | 12 |
| 40–49 | 8 |
| **Marital status** | |
| Never married | 5 |
| Divorced/ Widowed | 8 |
| (Re)Married | 13 |
| **HIV diagnosis** | |
| 1–5 years ago | 16 |
| 6–10 years ago | 7 |
| 11–15 years ago | 3 |
| **Religion** | |
| Islam | 23 |
| Catholic | 2 |
| Protestant | 1 |
| **Education** | |
| University graduate/Diploma | 6 |
| Senior High school graduate | 13 |
| Junior High school graduate | 6 |
| Elementary school graduate | 1 |
| **Occupation** | |
| Housewife | 10 |
| Entrepreneur | 3 |
| Private employees (tailor, shopkeeper, banker, laundress, NGO worker) | 10 |
| Sex worker | 1 |
| University student | 1 |
| Unemployed | 1 |

HIV impact, and their access to HIV care services. The interviews were conducted in a quiet, secured and confidential room. The place and times for the interviews were mutually agreed upon by both the participants and the field researcher. In regards to self-response of WLHIV towards living with HIV and negative HIV-related experiences they faced, interviews were focused on (i) understanding various psychological challenges or feelings and emotions they experienced following their HIV diagnosis; (ii) the reactions or treatments these women received from family members and other people around them once their HIV status was known; and (iii) how they responded to those challenges or impact. The duration of the interviews ranged from 35 to 87 minutes. Only the researcher and participant were present in the room during the interview, and none of the participants was known to the researcher prior to this study. No repeated interviews were conducted. The recruitment of participants and interviews ceased when the researchers felt that information provided by the participants had been rich enough to address the study objectives and data saturation has been reached.

## Data analysis

Audio recorded digitally, interviews were conducted in Bahasa Indonesia, the national language of Indonesia. Interview recordings were transcribed verbatim manually using a laptop

and for the purpose of publication, the quotes were translated into English. Transcripts were imported into NVivo 12 where the comprehensive analysis of the entire data was performed. Data analysis was guided by Ritchie and Spencer's qualitative data analysis framework [26]. This framework suggests five steps for qualitative data analysis: (i) familiarisation with the data, which was performed by reading each transcript repeatedly, breaking down the data into chunks of data, providing comments and labels to data to look for ideas, meanings and patterns; (ii) identification of a thematic framework which was undertaken by writing down key issues and concepts highlighted in the first step. They were then changed and refined throughout the analysis process and used to form a thematic framework; (iii) indexing the data by providing open codes to data extracts of each transcript. This was followed by close coding to identify similar or redundant codes to reduce the long list of codes to a manageable number. Codes that fell into the same theme or sub-theme were then grouped together; (iv) charting the data by rearranging or reorganising themes and their codes that had been created in previous steps in a summary chart to compare the data within each transcript or across the transcripts; and (v) mapping and interpretation of the data as a whole [26, 27].

## Ethical consideration

Ethics approvals for this study were obtained from the Social and Behavioural Research Ethics Committee, Flinders University (No. 8286), and the Health Research Ethics Committee, Duta Wacana Christian University, Yogyakarta, Indonesia (No. 1005/C.16/FK/2019). Prior to the interviews, each participant was informed about the purpose of the study and that the study had obtained ethical approvals. Study participants were advised about the voluntary nature of their participation and that they had the right to withdraw their participation at any time, without consequence. They were also advised that the interview would take approximately 45 to 90 min and would be recorded using a digital recorder. They were assured that the data or information that they provided during the interview was confidential and unidentifiable. Each interview transcript was assigned with a specific study identification letter and number to prevent the possibility of linking back the data or information to any individual in the future. Each participant received a reimbursement of IDR 100,000 (±USD 7) for transport and their time. Participants were provided with the written informed consent form and signed and returned it to the researcher before commencing the interviews.

## Results

The findings were grouped into two main themes, including (i) HIV-related challenges (e.g., psychological pressures, stigma and discrimination) WLHIV faced following their HIV diagnosis; and (ii) the women's self-response to HIV-related challenges facing them (e.g., searching for advice and support, pursuing strategies to rebuild trust and regain respect and acceptance from families, and educating themselves and their families about HIV). These are presented in detail below.

### HIV-related challenges WLHIV faced following the diagnosis

The HIV diagnosis was reported to cause a range of negative psychological challenges for the participants. The psychological challenges were experienced by these women at a certain point following their HIV diagnosis, and reflected in a range of negative feelings or emotions, such as feeling stressed, depressed, shocked, scared, broken down and desperate. The following quotes illustrate the feelings or emotions experienced by these women after their HIV diagnosis:

*"At that time (once diagnosed with HIV) I was shocked and very much stressed out. . . .. I cried every day because I was stressed out thinking of this (HIV). I felt like my future was blurred already. I felt broken. If I remember that moment I still cry"* (Participant 9, married).

*"The first time I was informed that I contracted HIV I felt broken, did not believe it. . . .. I thought I did not have a future anymore. That was in my mind. I thought I did not have hope for the future. I was very much depressed at that time"* (Participant 22, divorced).

Such psychological challenges seemed to be triggered by various negative thoughts and concerns that came into mind after their HIV diagnosis. For example, the fear of mother-to-child transmission (MTCT) was among the many, as supported by the following quote; '*I was frightened (of MTCT) because I gave birth normally and breastfed my child. I was so worried for several years before the (HIV) test'* (Participant 10 remarried). Fear of the possibility of shame for family members due to their HIV diagnosis was also another reason. For example, a participant said *"This (HIV status) can make them (her parents) feel ashamed if other people know because people must think that I have got it from sex while I am not married. Thinking of this makes me stressed out and depressed"* (Participant 19, single). Other examples of these were the thoughts and concerns about early death and its impact on the future of their children, and the possibility of being abandoned by their husbands due to their HIV-positive status:

*"What was in my mind was that I am living with HIV and would die soon. . . .. I thought I would die soon and then what about my child, what about the future of my child"* (Participant 6, remarried).

*"I was scared that my husband would leave me, but I had to tell him (about her HIV-positive status) because we have a child"* (Participant 5, married).

The HIV diagnosis was also reported by participants as causing stigma towards them by their own families and community members and healthcare professionals. These were reflected in others' negative assumptions or attitudes towards them. For example, they were being suspected as 'naughty women' or sex workers who have sex with multiple male partners, through which they acquired HIV infection:

*"I got the first negative treatment from my mom after I told her I was diagnosed with HIV. . . .. She asked, 'What have you done?'. . . .. I explained to my mom that I got HIV from my husband, it was not because I engaged in free sex or I was a sex worker, but my mom did not believe me, saying: 'You are just defending yourself'. She assumed that I was a 'naughty woman'"* (Participant 24, divorced).

*"I underwent a medical check-up, and the laboratory staff (a healthcare professional) asked me: 'how did you get it (HIV)?' I got it from my (late) husband, I said. 'Is your husband dead?' Yes, I replied. 'Did your husband like 'jajan' (having sex with female sex workers)?' In their mind, people who contracted HIV must be 'naughty' (a female sex worker). I got the same questions before: are you 'naughty'? Do you like jajan?'* (Participant 3, remarried).

HIV stigma towards participants was reported to also manifest in a range of discriminatory behaviours of other people. Participants described how they were treated negatively by their families, other community members and healthcare professionals following the disclosure of their HIV status. The discriminatory behaviours included separation of personal belongings, such as eating utensils, foods and rooms, from those of other family members, and separation of their children from them due to the fear of the spread of the infection. It was also reflected

in rejection, physical avoidance and the spread of HIV status by other community members towards women living with HIV:

*"She (her mother) took out all my clothes. I was asked to use my own plate and glass and wash them by myself. . . then my child was kept away from me. She said, 'Do not get close to your child, you may transmit it (HIV) to your child'. I was not allowed to touch my child for three months. Every time I got close to them my mother chased me away. I was put in a separate room. My food was given to me through the bottom of the door, just like you would do for a dog" (Participant 7, divorced).*

*"I got discrimination in the community where I lived before. If I have touched any foods, then people would not eat those foods. Some (community members) spread information that I am HIV-positive. I experienced this for about two years. Some avoided me, did not even want to shake hands with me" (Participant 17, divorced).*

HIV stigma against women living with HIV was also reflected in negative and refusal of treatment for them and their families living with HIV. The following narrative of a widow whose husband died from AIDS and child was denied health treatment after being diagnosed with HIV sums up the HIV stigma against them and their family members:

*"I need lots of courage just to come to the hospital. I feel traumatised with hospitals, and my body gets cold if I see hospitals. I was treated very badly by the doctor in the previous hospital. My child was not provided with the (antiretroviral) medicines due to the reason that there should be a healthy (HIV-negative) family member who accompanied her, otherwise the medicines would not be provided. Once a healthy family member of mine accompanied her (to access the medicines), the doctor said 'wait until her dad is fully recovered'. My husband was sick (living with HIV and hospitalised). My child who is positive living with HIV was not allowed to pup (use the toilet) in the hospital" (Participant 2, widowed).*

The narratives of these women illustrate strong psychological (fear, stress, depression and anxiety) and social challenges (stigma and discrimination) faced following their HIV diagnosis. These psychological challenges are caused by various negative concerns, thoughts that came into mind after the diagnosis, and HIV stigma and discrimination against them by their families, community members and healthcare professionals.

## The women's self-response to HIV-related challenges facing them

**Searching for advice and support.**   The narratives of the participants showed that even though they experienced stigma and discrimination by their family members, as well as various negative psychological pressures, they demonstrated strong resilience and capacity to find solutions to these challenges. They pursued effective strategies and support to protect themselves and regain their strengths. For example, they actively searched for advice and support from doctors, and peers who had been living with HIV for years which could enable them to accept their condition or HIV status and overcome the pressures and negative feelings facing them as also corroborated by the following quotes:

*"I went to the hospital with her (a companion of PLHIV from an NGO, who was also living with HIV). I got lots of information and support from her. . . .. So, I tried very hard to accept my condition and overcome the pressure I felt. I again met the counsellor at the hospital and told her that I was experiencing discrimination and that I was getting very mistreated by my*

*mom. I told her that I tried to convince my mom that this was not my fault so that my mom could accept me. . . .. I was also very active in a peer support group (comprising PLHIV, and their companions, doctors and nurses) activities and meetings, through which I met and shared with other friends who have lived with HIV for years and accepted their condition and looked healthy. Listening to their stories helped me a lot to overcome the psychological pressures I felt" (Participant 24, divorced).*

A similar story was echoed by another woman who was helped by her HIV-negative husband to search for advice and support from a HIV counsellor. This was acknowledged as helping her to cope with HIV stigma, discrimination and their negative consequences, including psychological pressure and stress she faced:

*"With support from my husband, I encouraged myself to meet a (HIV) counsellor (at the hospital) because I was so depressed due to the HIV diagnosis. I talked to the counsellor openly about my condition and asked her what I should do. I met the counsellor a few times because I was aware that I had to be strong to help myself. The counsellor gave me lots of advice, and the main points were that I have to accept myself and my condition, think positively, not get stressed, start the treatment, and routinely take medicine (ART). . . .. All the discriminatory and stigmatising attitudes and behaviours from my sisters-in-law and the extended family of my husband increased the pressure I felt. I was so stressed out because personally, I was still at the phase where I did not fully accept my own health condition (HIV status). But I kept on trying and consulted with the doctor and companions of PLHIV. I tried very hard, and luckily my husband supported and helped me to overcome the psychological pressures I experienced" (Participant 5, married).*

The participants' determination to focus on HIV treatment with antiretroviral therapy was also another positive response that demonstrated their resilience towards HIV-related psychological pressure and social challenges they faced. The narratives of a participant showed that even though they experienced tremendous pressure due to their own as well as their child's HIV diagnosis, they decided to fight against the burden and pressure they faced, and focus on the treatment of their ill-health condition. The following story of a divorced woman with a child who was also HIV-positive illustrates such experiences:

*"After my daughter and I were diagnosed with HIV, I kept silent for two years. I didn't talk to my parents and sisters. I didn't tell them about our HIV status. I felt stressed out and guilty. I was so depressed for two years and carried the burden by myself. . . .. One night my dad talked to me, very short, he asked me to surrender to God. I thought about it the whole night. I woke up early in the morning, encouraged myself, and brought my daughter to the priest to bless her. I told myself that whatever happens, I will accept it, the important thing is that I focused on the treatment for me and my daughter. Since then, I have really focused on treatment because I realised that I had to do something so that my daughter and I didn't die. My daughter and I adhere to the treatment up to now, we don't miss the treatment even once. Thank God, we are healthy" (Participant 16, divorced).*

Despite experiencing tremendous psychological challenges due to being diagnosed with HIV, these women demonstrated remarkable resilience and capacity to overcome the challenges. These were reflected in their strong effort and determination to pursue strategies and support from healthcare professionals and HIV counsellors, and focus on their treatment.

**Pursuing strategies to rebuild trust and regain respect and acceptance from families.**
The participants also reported having put in great efforts in pursuing effective strategies to help them rebuild trust and regain the respect and acceptance of their family members. They described how they tried to make their family members trust and accept them through some activities. These included taking their family members with them to consultation with HIV counsellors or medical doctors, and to peer support group meetings to listen to the perspectives of healthcare professionals and real-life experiences of PLHIV. These are reflected in the following narrative of a woman who acknowledged getting HIV from her spouse and was discriminatively treated by her mother:

*"They (the counsellor and doctor) suggested that I come to the hospital with my mom so that they could explain to her. I talked to my mom, saying: 'The doctor said that you and I should go together to the hospital tomorrow'. 'What for?' she asked..... I tried to ask her to go with me to the hospital. My mom kept on saying: 'Mom is getting bored, that was the reason you got beaten by your husband' (her mom accused her of engaging in free sex). I felt awry. . . .. My mom judged me (negatively), thinking that I am a 'naughty woman'. But I kept trying, and it took me a few weeks to finally get her agreement to get to the hospital with me. Finally, my mom and I went to the hospital, and the doctor and counsellor explained to her that I did not get HIV because I engaged in free sex or I was a 'naughty woman'. The counsellor also told her: 'Madam, do not be afraid to touch your daughter, eat together or wash your clothes together, it is fine'. After she was told by the counsellor at the hospital and knew the information about HIV and its treatment, she started to care about me, reminded me to take the medicine (ART). . . .." (Participant 24, divorced).*

A similar story was echoed by several women who received negative treatments from their in-laws due to their HIV-positive status. However, they responded to such treatments by seeking support from other family members or companions of PLHIV, which led to them regaining the acceptance of their in-laws. This effort showed their resilience towards the HIV-related challenging situation facing them after their HIV diagnosis. The following quote of a widowed who was rejected by her in-laws after the death of her spouse clearly illustrates such effort and resilience:

*"My sister-in-law seemed so scared of this disease (HIV infection). She asked us (the woman and her daughter) to go back to my parents (after her HIV diagnosis and the death of her husband). We didn't come to my parents after the death of my husband, we stayed there (in Kalimantan) for two years. At the time there was a companion of PLHIV who was so kind to me and my daughter every time we met. I asked her to come to our house and talk to my in-laws about HIV. She came to visit us and tell them that HIV is not easily spread through social interaction, eating utensils or clothes. After that, they started to step-by-step close to us, care about us, and allow their children to play with my daughter. I am happy because they accept us and we still have good relations until now, we still contact each other even though my daughter and I came back here (Yogyakarta) and live with my parents" (Participant 11, widowed).*

These women's narratives about helping family members to gain information and knowledge about HIV-related issues as a strategy to rebuild broken trust and relationships with families caused by their HIV diagnosis also illustrated their strong resilience and capacity in the HIV-related challenging situation. This effort helped them regain acceptance and greater support from families which seemed to be a boost for them psychologically and socially to overcome the HIV impacts they faced.

## Educating themselves and family members about HIV

Educating themselves about HIV and its impact was another initiative taken by some participants to overcome psychological pressure and negative thoughts that came into mind following their HIV diagnosis. For example, having a better understanding of HIV and its impact helped them cope with the fear of death and transmission to their family members, especially their children:

> "…. I was so scared that I would die soon and every time I touched my children, I always had the thought in mind that they could get the virus from me. I felt pressured. That was the reason why I started searching on the internet and reading about HIV and was regularly involved in peer support group meetings. After acquiring much information about this infection from what I read and peers I started to feel calm …." (Participant 23, single).

Similarly, the participants took the initiative to educate their family members regarding the HIV issue or how HIV is transmitted. This was done through direct dissemination of information about HIV to family members and the provision of pamphlets containing HIV information. These were used as strategies to increase family members' knowledge and understanding of HIV, regain their respect and acceptance and reduce their discriminatory and stigmatising attitudes and behaviours towards the participants:

> "I also tried very hard to make them accept me again. My husband and I searched for information about HIV and talked to the doctor about how the infection spreads. Then my husband and I explained to his sisters step by step about how HIV spreads and that HIV is not transmitted through social contacts in daily life, it spreads through blood and body fluids. Finally, step by step they accepted me and did not separate my eating utensils. After we had our second child, who was HIV-negative because we had planned and consulted with the doctor since the very beginning (of her pregnancy), I invited the entire extended family of my husband to celebrate the second birthday of my child in my house. At that time, I told them that my child was HIV negative. Since that time, the extended family of my husband became aware that HIV does not easily spread, and they started to accept me step by step" (Participant 5, married).

> "After my mom and I met the counsellor and doctor, I continued to search for information about HIV which I could give to her to read. So, I often took pamphlets from the HIV clinic I visited and from XX (the name of a non-governmental organisation that provides support for PLHIV) and brought them back home for my mom to read. She read every single pamphlet I gave her and often discussed with me things related to HIV that she read. So, it was very helpful for her and changed her views about me, HIV and people who are infected with HIV in general. I am glad that finally, my mom shows her love again to me" (Participant 21, widowed).

These women suffered stigma and discrimination from their family members, which also caused psychological pressures on them. But in such challenging circumstances, they took on more responsibility for their own lives and tried to turn the circumstances in their favour by educating their family members about HIV. This not only led to them regaining respect and acceptance, but also support from their family members, which was very helpful for them to cope with the difficulties of living with HIV.

## Discussion

This paper focuses on understanding the HIV-related experience of WLHIV in Yogyakarta, Indonesia, to deeply explore the self-response and capacity they employ to cope with the difficulties of living with HIV. The narratives of WLHIV in this study assist us in understanding how they make sense of their lived experience of HIV. In contrast to much of the previous HIV literature that does not take into account women's self-response and capacity to seek support, and portrays women as subordinate, submissive, victimised, tradition-bound, and ignorant [4, 28–30], our study presents WLHIV as actively engaged in their environments so they can change their circumstances surrounding their HIV diagnosis. It shows these women's capacity and response through significant efforts to overcome HIV-associated psychological pressures, stigma and discrimination facing them, and to convince their family members that their HIV status is not their fault, nor anything to be ashamed about. It also shows these women's great resilience to try and rebuild and nurture relationships with their families after the discriminatory behaviours had ceased.

Previous studies have reported that WLHIV experience significant psychological challenges, such as feeling stressed, depressed, angry, guilty, and self-blaming due to their HIV diagnosis [4, 6, 8]. Other reported strong supporting factors for stress, depression and anxiety among WLHIV are stigma and discrimination they faced from family members, including spouses, parents, in-laws and relatives, community members and HCPs [4, 8, 12, 13]. Similar findings are also apparent in the narratives of WLHIV in our study, which highlighted their experiences of negative feelings and emotions such as fear, shame, stress, anxiety, depression, and stigma and discrimination including negative labelling, abusive comments, isolation, refusal, rejection and separation of personal belongings by others. These discriminatory behaviours are stigma mechanisms reflecting psychological responses of non-infected people who hold negative beliefs that the women may transmit the virus to them or threaten their life and health [31, 32]. Thus, the status of living with HIV and these negative labels, which are reflected in negative emotions or feelings (often anger and fear) of their family members or other people towards them, are the devalued attributes attached to these women [31, 33].

Findings of previous studies have reported on the topic of the resilience of WLHIV towards challenging circumstances of living with HIV. For example, several studies report supporting factors for WLHIV's resilience such as employment, marital, family and social relationships, social support, community engagement and absence of stigma and discrimination [34–38]. Other studies have also reported on optimism, will to live, higher self-efficacy and self-esteem, and lower traumatic and depressive symptoms as factors that support the resilience of WLHIV [39, 40]. What our data add to the existing literature is that despite experiencing a range of psychological challenges, stigma and discrimination, and lack of family and social support, these women demonstrated impressive self-response and capacity to seek support and treatment, and pursue effective strategies to overcome or change their circumstances to more positive ones. For example, they tried to accept their HIV status, focus on their treatment and overcome the psychological challenges facing them, and at the same time convince their family members who were accusing and discriminating against them, that their HIV status was not their fault. Their capacity and response to the challenging situations were reflected in the strategies they used, including consulting an HIV counsellor or medical doctor, HIV-positive peers, companions of PLHIV, and initiating treatment, which led them to step by step overcoming significant psychological pressures they faced. Their strong efforts to regain acceptance from and re-establish a positive relationship with their family members based on care, love and support, also reflected these women's resilience towards HIV-related challenging circumstances facing them. Similarly, the women's determination and great deal of persistence to

seek information and invest time in teaching and educating their family members also reflected their capacity and self-response towards HIV-related challenging situations they faced.

Our findings showed the women's incredible capacity for resilience and self-preservation in the face of extreme psychological pressure, discrimination and hurt from their own families and others. They turned what was a very difficult and painful situation into an opportunity to pursue external support from counsellors, doctors and companions of PLHIV, and more remarkably, dedicated time and resources to encourage their families to educate themselves, so the women felt accepted and supported. In addition, it should be acknowledged that the availability and accessibility of HIV-related services and support in healthcare facilities and non-governmental organisations concerned with HIV issues, and from healthcare professionals, companions of PLHIV and peer support groups of PLHIV in Yogyakarta [23, 41, 42], seemed to play important role in the women's successful response to HIV-related challenging circumstances they faced.

## Study limitations and strengths

Our study has some limitations that need to be considered in interpreting its findings. The use of the snowball sampling technique for the recruitment of the participants seemed to have led to the inclusion of participants from the same networks as the current participants. Also, the initial recruitment point through an HIV clinic seemed to have led to the recruitment of WLHIV who had already been on ART and had been coping with their HIV-positive status relatively well. These might have led to us under-sampling WLHIV from outside of the current participants' networks and who had not yet accessed HIV care services, who may have different stories to share. However, the strengths of our study are that it is an initial study that explored the self-response and capacity of WLHIV to pursue effective strategies and support during the challenging circumstances following their HIV diagnosis. Our findings have strong implications for both health and non-health sectors in Yogyakarta, Indonesia and other similar settings. WLHIV experience detrimental impacts of HIV, thus there is a need to address their needs through policy and practice to help them cope with HIV-related psychological and social challenges effectively. Family and community members seem to play an important role in stigma and discrimination against WLHIV, thus there is also a need for HIV education programs for them to enhance their HIV-health literacy and acceptance of PLHIV.

## Conclusion

This study reveals great self-response and capacity of WLHIV in this study in not only coming to terms with their own diagnosis, but also in their response to accessing treatment and support, and perhaps most importantly, in their determination to educate family members, so they felt acceptance and love. They were very articulate in their understanding of the challenging circumstances and were tireless in finding solutions to these circumstances, including encouraging family members to talk to HIV counsellors and doctors about HIV, searching for HIV information and explaining it to family members to make them understand. These women demonstrated that they were not ignorant or passive after their HIV diagnosis, but actively pursued effective strategies to overcome HIV-associated negative challenges facing them. The findings indicate the importance of developing evidence-based intervention programs that address the needs of WLHIV and increase their resilience and capacity to access available supports. Future studies should continue to further unpack this idea of self-response and capacity of women diagnosed and living with HIV, especially in the face of significant psychological and social challenges. There is also a need to explore the role of governmental and non-governmental institutions in enabling WLHIV to empower themselves.

## Supporting information

**S1 File.**
(DOCX)

## Acknowledgments

We would like to thank the participants who had voluntarily spent their time participating in this study and provided us with valuable information.

## Author Contributions

**Conceptualization:** Nelsensius Klau Fauk, Hailay Abrha Gesesew, Lillian Mwanri, Karen Hawke, Paul Russell Ward.

**Formal analysis:** Nelsensius Klau Fauk.

**Methodology:** Nelsensius Klau Fauk, Hailay Abrha Gesesew, Lillian Mwanri, Karen Hawke, Paul Russell Ward.

**Project administration:** Nelsensius Klau Fauk.

**Writing – original draft:** Nelsensius Klau Fauk.

**Writing – review & editing:** Nelsensius Klau Fauk, Hailay Abrha Gesesew, Lillian Mwanri, Karen Hawke, Paul Russell Ward.

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
