## [Decision Letter · Decision Letter 0]

29 Jun 2022

PONE-D-22-13562HIV-related challenges and women’s self-response: a qualitative study with women living with HIV in IndonesiaPLOS ONE

Dear Dr. Gesesew,

Thank you for submitting your manuscript to PLOS ONE. After careful consideration, we feel that it has merit but does not fully meet PLOS ONE’s publication criteria as it currently stands. Therefore, we invite you to submit a revised version of the manuscript that addresses the points raised during the review process.

We look forward to receiving your revised manuscript.

Kind regards,

Carlos Alberto Zúniga-González, Ph.D

Academic Editor

PLOS ONE

Journal Requirements:

3. Please amend your current ethics statement to address the following concerns:

a) Did participants provide their written or verbal informed consent to participate in this study?

Additional Editor Comments:

Dear author your article is a good contribution to women with problems HIV, However is necessary to make minor improvements. I suggest you focused in methodology used. I suggest some reference that to aid

[1] Zúniga-González, C. A., Jarquín-Saez, M. R., Martinez-Andrades, E., & Rivas-Garcia, J. A. (2016). Investigación acción participativa: Un enfoque de generación del conocimiento. Rev. Iberoam. Bioecon. Cambio Clim., 2(1), 218–224. https://doi.org/10.5377/ribcc.v2i1.5696

[2] Rios-Molina, L. M. (2016). Mujeres y participación política en Nicaragua, 1980 -2015. Rev. Iberoam. Bioecon. Cambio Clim., 2(4), 556–562. https://doi.org/10.5377/ribcc.v2i4.5930

Reviewers' comments:

Reviewer's Responses to Questions

**Comments to the Author**

1. Is the manuscript technically sound, and do the data support the conclusions?

Reviewer #1: Yes

Reviewer #2: Yes

2. Has the statistical analysis been performed appropriately and rigorously? 

Reviewer #1: N/A

Reviewer #2: Yes

3. Have the authors made all data underlying the findings in their manuscript fully available?

Reviewer #1: No

Reviewer #2: Yes

4. Is the manuscript presented in an intelligible fashion and written in standard English?

Reviewer #1: Yes

Reviewer #2: Yes

5. Review Comments to the Author

Reviewer #1: � As authors are required to make all data underlying the findings fully available without restrictions; please try and describe the data availability and the restrictions in detail. We understand that the authors are not willing to share the data due to its sensitive nature and keeping the identities of the study participants. However, please explain and justify why the data cannot be shared while maintaining anonymity or by using pseudonym for keeping the study participant’s identity secret.

Please move the ‘Characteristics of the participants’ from the results to the ‘Methods’ section

It’s a good idea if the participants can be demonstrated in a table with some basic dynamics

It’s a good idea to give a bit of narrative in between the quotations. Three or four quotations together is a bit tiring. Try to establish your argument using the quotations and put some narrative in between,

Collate typical and atypical responses and discuss accordingly

At the end of a section/paragraph, try to summarize the section in a couple of lines with some conclusive remarks. Rather than abruptly finishing a section with a couple of quotations.

Please double-check your references and bibliography

Please do the formatting of the manuscript properly.

Reviewer #2: Comments: HIV-related challenges and women’s self-response: a qualitative study with women living with HIV in Indonesia (PONE-D-22-13562)

This is an interesting paper. I have some comments as follows:

• Please explain what are PLHIV and WLHIV stands for in abstract as well as in the paper for the readers.

• Make sure you write succinctly – there are many sentences which are too long. For instance, on page 5, authors write: “It is crucial to understand the lived experience of WLHIV themselves in order to effectively address the impact of the HIV epidemic at multiple levels, from immediate healthcare and mental health support to developing evidence-based programs and interventions that address the needs of WLHIV and increase their resilience and capacity to access these supports, and to driving government legislation and policy change.”

• What is CD4 on page 5?

• Need to explain why authors collect data from “Yogyakarta city” only.

• Correct the grammar. For example:

o On page 6, authors write: “One-on-one in-depth interviews were used to explore participants’ perceptions of factors leading to HIV infection, their lived experienced of HIV impact and of their access HIV care services.”

o On page 7, authors write: This was followed by close coding to identify similar or redundant codes in order to reduce the ling list of code to a manageable number.

o On page 10, authors write: These were reflected in others’ negative assumptions or attitudes towards them. For example, they were being suspected as a ‘naughty woman’ (means a sex worker or a woman who has sex multiple men), through which they acquired HIV infection:

There are others.

• On page 8, what do you mean by “single?” – is it never married?

• I wonder whether women’s and men’s HIV related experience are different? If so, how? I understand, authors only collected data from women, but if authors can explain or motivate on why only women, or how women’s experience can be different than men, that will be a good addition to the paper.

6. PLOS authors have the option to publish the peer review history of their article (what does this mean?). If published, this will include your full peer review and any attached files.

Reviewer #1: **Yes: **Raafat Hassan

Reviewer #2: No

---

## [Author Response · Author response to Decision Letter 0]

21 Aug 2022

A document detailing responses to the Editor is attached.

---

## [Decision Letter · Decision Letter 1]

15 Sep 2022

HIV-related challenges and women’s self-response: a qualitative study with women living with HIV in Indonesia

PONE-D-22-13562R1

Dear Dr. Gesesew,

We’re pleased to inform you that your manuscript has been judged scientifically suitable for publication and will be formally accepted for publication once it meets all outstanding technical requirements.

Kind regards,

Zuardin Zuardin, Ph.D

Academic Editor

PLOS ONE

Additional Editor Comments (optional):

Reviewers' comments:

Reviewer's Responses to Questions

**Comments to the Author**

1. If the authors have adequately addressed your comments raised in a previous round of review and you feel that this manuscript is now acceptable for publication, you may indicate that here to bypass the “Comments to the Author” section, enter your conflict of interest statement in the “Confidential to Editor” section, and submit your "Accept" recommendation.

Reviewer #1: All comments have been addressed

2. Is the manuscript technically sound, and do the data support the conclusions?

Reviewer #1: Yes

3. Has the statistical analysis been performed appropriately and rigorously? 

Reviewer #1: N/A

4. Have the authors made all data underlying the findings in their manuscript fully available?

Reviewer #1: No

5. Is the manuscript presented in an intelligible fashion and written in standard English?

Reviewer #1: Yes

6. Review Comments to the Author

Reviewer #1: Thank you very much for addressing all the comments succinctly and accordingly. This is great. The reason behind not making all the data available was explained. Appreciate all your hard work.

7. PLOS authors have the option to publish the peer review history of their article (what does this mean?). If published, this will include your full peer review and any attached files.

Reviewer #1: **Yes: **Raafat Hassan

---

## [Editor Report · Acceptance letter]

29 Sep 2022

PONE-D-22-13562R1 

HIV-related challenges and women’s self-response: a qualitative study with women living with HIV in Indonesia 

Dear Dr. Gesesew:

I'm pleased to inform you that your manuscript has been deemed suitable for publication in PLOS ONE. Congratulations! Your manuscript is now with our production department. 

Kind regards, 

on behalf of

Dr Zuardin Zuardin 

Academic Editor

PLOS ONE